# "Without a man's decision, nothing works": Building resilience to Rift Valley fever in pastoralist communities in Isiolo Kenya

**Irene N. Mutambo**[1,2]*, **Bernard Bett**[2], **Salome A. Bukachi**[1,3]

1 Institute of Anthropology, Gender and African Studies, University of Nairobi, Nairobi, Kenya, 2 One Health Research, Education and Outreach Centre in Africa, International Livestock Research Institute, Nairobi, Kenya, 3 Department of Anthropology, Durban University, Durban, United Kingdom

* mutamboirene@yahoo.com

## Abstract

Rift Valley Fever (RVF) is a zoonotic disease that affects both livestock and humans. Men and women in pastoralist communities are vulnerable to RVF risk exposure because of their different roles and reliance on livestock products. This study sought to understand how ownership and decision-making in pastoralist male and female-headed households influence coping mechanisms and resilience to Rift Valley fever (RVF), using the three resilience capacities of absorptive, adaptive, and transformative. This study was conducted in two sub-counties (Garbatulla) and Merti), Isiolo County, Kenya. Data were collected through 16 focus group discussions and 13 key informant interviews with pastoralists and animal and human health stakeholders. The findings indicate that traditionally, men have the final say on decisions related to livestock ownership and make overall household decisions. Pastoralist men and women employ different approaches, including hygiene practices and mosquito nets, community knowledge dissemination, establishment of new businesses, utilization of healthcare, and indigenous medicines, to reduce the effects of RVF in both humans and livestock. They also collaborated with community disease surveillance initiatives to strengthen disease surveillance networks and gain access to county government support. This process fosters resilience, community empowerment, and transformative and sustainable adaptation responses to RVF.

**Data Availability Statement:** All relevant data are within the paper and its Supporting Information files.

## Introduction

### Rift Valley fever

Rift Valley fever (RVF) is a zoonotic viral disease that can be transmitted from animals to humans [1,2] by infected *Aedes* and *Culex* mosquitoes. Outbreaks typically occur after heavy rainfall, affecting livestock, such as sheep, goats, cattle, buffaloes, and camels, which serve as primary reservoirs for human infection [3–5]. The disease mostly affects the pastoralist population, who are reliant on livestock for food and livelihood [5–7]. Humans contract RVF through direct and indirect exposure. Direct exposure includes consuming raw or

**Funding:** This work was supported by the One Health Research, Education and Outreach Centre in Africa funded by The Federal Ministry of Economic Cooperation and Development (BMZ) and led by International Livestock Research Institute, Nairobi, Kenya. The funders had no role in study design, data collection and analysis, decision to publish, or preparation of the manuscript.

**Competing interests:** There are no competing interests by authors

undercooked animal products (e.g., milk, blood, and meat), handling infected animal bodily fluids, and inhalation of aerosols from infected environments [8–10]. Indirect exposure occurs when people come in contact with infected vectors [11,12]. RVF poses a significant threat to pastoralists and their livestock, compromising their ability to meet basic needs and respond to outbreaks [5,13]. RVF was first reported in the Rift Valley region of Kenya in 1931 [14]. Since its initial detection, Kenya has experienced several outbreaks, notably in 1998, 2006/7, 2014, 2018, 2020, and 2024, affecting various parts of the country [14–16]. The 2020 outbreak particularly affected Isiolo and Mandera counties, with approximately 32 human cases and 11 deaths recorded in early February [15].

A focus on decision-making power dynamics in male- and female-headed households from a gender perspective presents an important opportunity to understand the different gender roles and relations regarding adaptation strategies and innovations [17,18]. For instance, men are often the primary final decision-makers on aspects such as the sale of livestock, access to health, household expenditures, and access to information/participation, while women are often involved in making decisions about household chores such as nutrition and children's health [19,20]. This division limits women's ability to adapt to crises such as RVF outbreaks. Some studies on the adaptation of agricultural innovations suggest that joint decision-making in households strengthens the capacity of men and women to cope with challenges, highlighting the importance of collaboration between them during such events [21]. In addition, pastoralist communities are often rooted in patriarchal systems, concentrating on men's decision-making power [22]. This limits women's ability to control resources such as livestock, which are crucial assets that can help women cope better and negotiate for fallback during shocks such as RVF [23–25]. Studies have shown that even when women own livestock, they typically have fewer and less valuable animals than men. This limited ownership hinders their capacity to adapt and respond effectively to challenges, such as RVF outbreaks [20,24]. Similarly, although some men and women are knowledgeable about livestock diseases, both men and women in pastoral communities have limited access to crucial information on zoonotic diseases, such as RVF, especially regarding transmission and preventative measures [24,26]. This knowledge gap is widened by interventions that focus solely on men, thus allowing men and women to face difficulties in accessing effective strategies to control zoonotic diseases, such as RVF, which in turn challenges pastoralists' adaptive capacities to move beyond the vulnerability thresholds [25,27,28]. Importantly, because pastoralists reside in rural areas, they also face the challenge of poor infrastructure, such as roads, access to veterinary services, well-stocked hospitals, and timely surveillance, which weakens their ability to control outbreaks and build resilience [24,25].

Social-ecological resilience theory focuses on three capacities—absorptive, adaptive, and transformative—that people employ to recover from shocks and stresses within their social-ecological systems [23]. Absorptive capacities refer to the immediate measures and strategies pastoralist men and women use to withstand shocks. They aim not to lose essential functions but rather to minimize exposure to hazards and build buffers to absorb the effects of RVF. An example of absorptive capacity for a flooding community includes flood-resistant housing and early warning systems for natural disasters to reduce the effects of shocks [28]. Adaptive capacity refers to the ability of pastoralists to learn from experiences, adjust their behaviours, and adopt new strategies and innovations to manage future shocks [27]. For instance, adapting to new technologies to enhance livelihoods and respond to changing conditions. Finally, transformative capacity refers to the ability to create a structural atmosphere to change the barriers that hinder people from moving beyond vulnerability thresholds [29], such as the capacity to advocate for policy changes for equitable access to resources.

This qualitative study on how household decision-making and ownership influence resilience and gendered strategies adopted during RVF outbreaks provides information on how gender dynamics shape and influence men and women's capacities that enable negotiation for resilience during RVF outbreaks. This study aimed to understand how gender relations influence the resilience of pastoralist men and women to RVF in Isiolo County, Kenya. Our qualitative analysis focuses on how household ownership and decision making influence men's and women's resilience to RVF and the gendered adaptation strategies used to respond to the adverse impact of RVF outbreaks. The findings of this study on gender dynamics and resilience to RVF within pastoralist communities suggest a paradigm shift in approaching different coping strategies to reduce the effects of RVF. Future research on RVF should move beyond a one-size-fits-all approach and incorporate a gendered lens to design interventions that are more culturally appropriate and empower both men and women to build resilience in the face of RVF. This holistic approach has the potential to improve preparedness and response strategies for RVF outbreaks in pastoralist societies significantly.

## Materials and methods

This study investigates how gender relations influence the resilience of pastoralist men and women to RVF in Isiolo County, Kenya. This study examined household decision-making dynamics, where men typically have authority over significant matters while women manage domestic decisions. Utilizing social-ecological resilience theory, this research explores absorptive, adaptive, and transformative capacities to understand how these gendered relations affect coping and adaptation strategies during an RVF outbreak. The findings reveal how household decision-making and livestock ownership influence resilience, highlighting the gendered strategies employed in response to the disease.

### Study area

This qualitative study was conducted in the Garbatulla and Merti sub-counties of Isiolo County (Fig 1). Isiolo County has a population of 268,002 (128,483 female and 139,510 male) with an average household size of 4.6 members [30]. Permanent housing is concentrated in Isiolo Town, whereas rural areas predominantly feature temporary structures. The county is classified as arid or semi-arid, with an average annual rainfall of 400–650 mm [31]. This dry climate makes livestock the economic mainstay, supporting over 80% of the residents [31].

### Study design

This study employed a cross-sectional qualitative research design, using focus group discussions (FGDs) and interviews with key informants' interviews (KIIs). to explore ownership and household decision-making processes related to the (RVF) in Isiolo County. A cross-sectional design was appropriate for this study, as it allowed for a snapshot of decision-making practices and ownership at a specific point in time, providing valuable insights into the current landscape [32]). Qualitative research methods are effective for understanding participants' personal experiences, beliefs, and perspectives. By using focus groups and KIIs interviews, researchers gained a comprehensive understanding of the research topic. This combined approach enhances the depth and validity of the research findings through triangulation.

### Sample size and sampling approaches

This study involved a sample size of 129 participants, including 16 FGDs (involving 116 participants) and 13 key informants (KIs).This study recruited 16 focus group discussions to

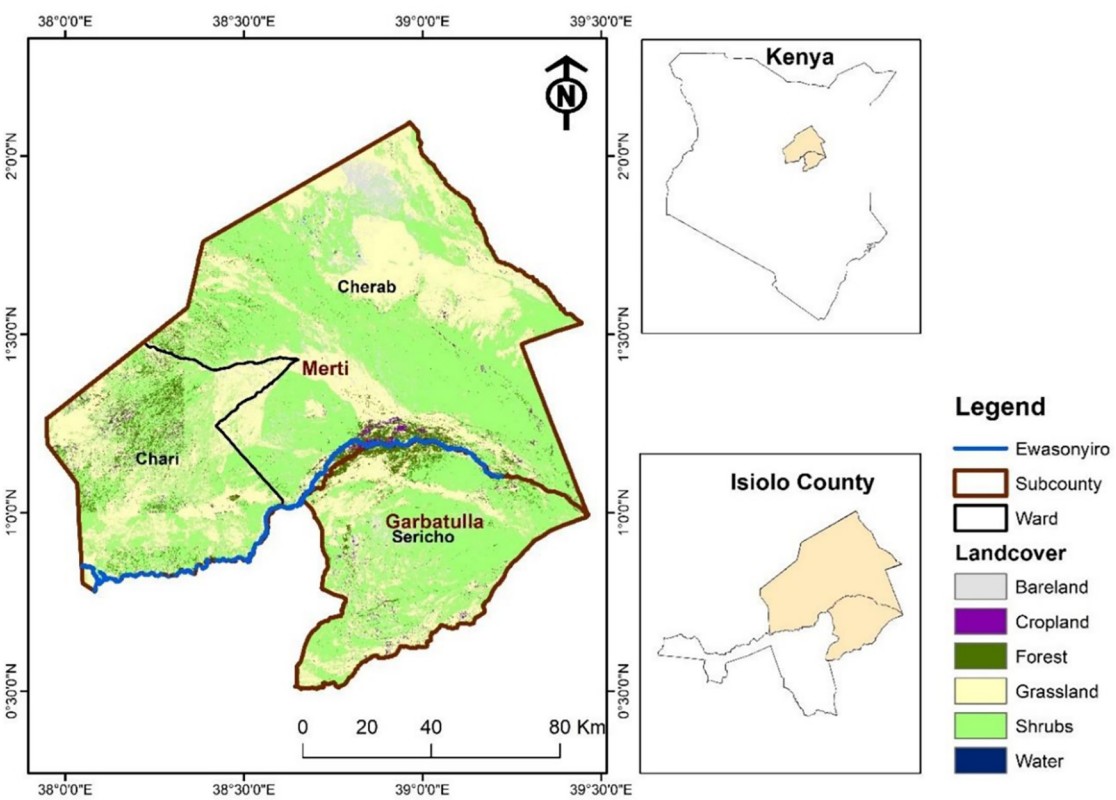

## Source: ILRI-GIS

**Fig 1. Map showing Garbutulla and Merti Sub-County.**

ensure adequate representation of each participant category (men and women) and to enable cross-group analysis. This decision adheres to the recommendation of qualitative data approach to conduct three or four focus group discussions per category [33]. By conducting 4 focus group discussions per category (per location), the study aimed to achieve a balance between data saturation and, ensuring that sufficient data was collected to reach a comprehensive understanding of the research objective. This approach also facilitated meaningful comparisons and insights across gender. To gather in-depth qualitative data, purposive sampling was used to select and ensure the inclusion of participants with relevant experiences who were knowledgeable and influential in RVF management in Isiolo County. The selection criteria required FGDs participants to be actively involved in herding livestock (sheep, goats, and/or cattle) and to have either personal or familial experience with RVF. To identify and recruit suitable participants who met these criteria, the researchers collaborated with community leaders familiar with local herders and their history of RVF. The KIs were purposefully selected for their expertise and experience with the Rift Valley Fever (RVF) in Isiolo County, as well as their diverse roles in RVF management and deep understanding of the local context.

A total of 116 participants (48 men and 68 women) participated in 16 FGDs (8) Merti and Garbatulla). Merti and Garbutulla Sub Counties were purposefully chosen for the study due to their history of Rift Valley Fever (RVF) outbreaks in the past. To ensure balanced gender representation without dominance by either gender, four separate FGDs were held with men

and women in each location. This included 60 participants (26 men and 34 women) in Merti, and 56 participants (22 men and 34 women) in Garbatulla. Across both locations, Each FGD comprised 8 to 12 participants, and included male herders, men from male-headed households, female herders, women from male-headed households, and women from female-headed households. This diverse representation ensured a variety of perspectives within and across each FGD.

## Ethical statement

Ethical approval and research permits were obtained from the relevant authorities in Kenya: the International Livestock Research Institute ethics committee (ILRI-IREC2022-56) and the National Council of Science, Technology, and Innovation (NACOSTI), Kenya (NACOSTI/P/23/2313). Others included approval from the Isiolo County Department of Veterinary and Local Area Chiefs. Informed consent was obtained from all participants after a thorough explanation of the research objectives, data collection methods, and potential risks and benefits, either through a signature or thumbprint. Anonymity and confidentiality of participant information were maintained throughout the research process.

## Data collection

Data collection involved two primary methods, including Focus Group Discussions (FGDs) and key interviews (KIIs). Discussions were facilitated by a moderator and recorded by note-takers and the participants were seated in a semi-circular arrangement.

Participants were asked open-ended questions about household decision making, adaptive strategies used to reduce the effects of RVF, and the existing structure for response. Vignettes were used in the FGDs to explore the household decision-making processes. The participants were given three colored cards to indicate who made each decision: orange for decisions made by men, red for decisions made by women, and green for joint decisions. Household decisions on the livestock market, access to health, household expenditures, and participation in or access to information were discussed.

The participants used cards to show how decisions were made in their households during the discussion. In separate discussions between men and women, participants used the cards to show how decisions were made in their households during the discussion. To prevent participants from being influenced by others, researchers counted from one to three, prompting participants to hold up their responses to a vignette question simultaneously. This simultaneous display of cards helped minimize bias, ensuring that the participants did not alter their initial responses. After each decision was indicated, the number of cards for each color was recorded, and the participants were asked to elaborate on their choices. The same activity was repeated after participants discussed the management strategies, they utilized to respond to RVF outbreaks. This interactive approach helped to reveal the gender dynamics and power structures within households. All group discussions lasted between 60 and 90 minutes and were held in the village either in the morning or afternoon, under a tree or shed.

We conducted KIIs with male key informants (KIs) and included officials from the veterinary and public health sectors, who are key stakeholders in Rift Valley Fever management in Isiolo County. Their insights were valuable because of their diverse roles in RVF management and deep understanding of the local context. Each interview lasted for 45 minutes and included questions on the signs of RVF in humans and livestock, management strategies to reduce the effects of RVFs, and factors that influence men's and women's vulnerability to RVF. Both the FGDs and KIIs involved audio recordings, and detailed notes were taken to supplement the recordings. Data were collected between March and April 2023.

### Data management and analysis

Data from FGDs and KIIs were transcribed verbatim from Borana to English by native speakers. The transcripts were subjected to rigorous quality assurance checks to ensure their accuracy and consistency. To enhance the reliability, axial coding was employed to identify patterns and relationships within the data, resulting in the development of a comprehensive coding framework. Inductive thematic analysis, grounded in the data and guided by this study's objectives, was conducted using NVivo 14. NVivo software facilitated the data coding, organization, and identification of key themes. The findings were reported with representative quotes to showcase diverse perspectives and enhance understanding. The authors held weekly meetings to discuss and interpret the emerging findings collectively. This served to reduce the biases of individual analysts and helped with triangulation and interpretation of the findings with an expert lens. Triangulation of data from KIIs and FGDs, including a comparative analysis of responses from men and women to identify similarities and differences, was conducted. This enhanced the validity and credibility of the findings.

## Results

### Livestock ownership and control

The study findings revealed that all the participants owned various livestock, including cows, sheep, goats, chickens, camels, and donkeys. Men were reported to own 65% of the livestock, and women owned 35% respectively. Specifically, out of the total number of animals reported, men owned (95%) cows, (70%) goats, (67%) sheep, (100%) camels (35%) donkeys. While Women were reported to own chicken (99%), donkeys (65%), goats (30%), sheep (33%), camels (0%), and (5%) cow. Gender disparities in livestock owners are associated with cultural norms and traditions (e.g., inheritance rights and household heads). Gender differences in livestock ownership between men and women were observed in the two study areas.

All FGDs revealed that women rarely inherit livestock (cows, goats, and camels) because of a lack of inheritance rights. This was confirmed by the local KIIs who stated that women acquire livestock through gifts from their parents as appreciation for their labour (goats, sheep, and chickens) or as a dowry upon marriage. There was consensus from the men and women groups that husbands also gifted livestock (donkeys) to women to help with chores, including fetching water and firewood. However, all women reported that men often claim gifted animals, limiting their decision-making power. Additionally, women revealed that if their husbands lost livestock due to predation, illness, or death, they often took their wives' animals as compensation for their losses. This is illustrated in the following narration:

> *"Women own livestock through dowry but men sell women's livestock without their knowledge. In addition, if men's livestock are taken away by bandits, or men come back home and take women's livestock. Men also take women's livestock if their animals fall sick and die or when the animals get lost during grazing for replacement" (Female herders FGD 02).*

### Absorptive resilience strategies and decision making

The group discussions with men, women, and KIIs revealed that during an RVF outbreak, pastoralists use mosquito nets, repellents, and hygiene practices as absorptive strategies to reduce and prevent the extreme effects of an RVF outbreak, as detailed below.

Men and women in all groups reported that, at the initial stages of an RVF outbreak, they used mosquito nets and repellents as an immediate adaptation strategy to reduce the effects of

RVF exposure in humans. Mosquito nets and repellents were mentioned as means to help protect households from mosquito bites and reduce the risk of RVF infection. Our findings also revealed that the decision to use mosquito nets and repellents within the household was made by the woman (wife), while the decision to purchase mosquito nets and repellents was made by the men (husbands). In addition, men were mainly reported to spray the household and kraal to chase and kill mosquitoes, and the decision to spray was made by men. Men decide to purchase mosquito nets, repellents, and pesticides because they own resources, are heads of households, and are viewed as presidents within the household.

> *"We sleep under mosquito nets and use repellents to avoid mosquito bites. Men are responsible for the purchase of mosquito nets and repellents because men are the household heads while women ensure that every member sleeps in a net" (Men from male-headed household FGD 01)*

KIIs and FGDs reported good hygiene practices as an important and effective response strategy to mitigate the effect of RVF. Hygiene practices reported included milking, boiling milk and meat, clearing bushes, setting fire/smoking in the livestock shade, and using gloves to support during birth. Our findings further revealed that during an RVF outbreak, members of each household must collaborate to implement hygiene practices to minimize the risk of exposure to the virus. All the FGD and KII participants' reasons for observing hygiene practices were linked to their effective preventive measures for maintaining the health and well-being of their families and the community. Participants in all FGDs highlighted that decisions about milking, fire tending, and bush clearing were reported to be made jointly between spouses based on who is present at home. However, although hygiene practices were reported as a collaborative effort, all participants agreed that the women took the lead in boiling milk and meat, providing home-based care, and discarding broth after boiling meat suspected to be from RVF-infected animals. Men, in turn, have been reported to provide crucial support by collecting firewood and lighting fires. Notably, the decision to slaughter an animal was reported to belong to the men by all FGD participants, and the decision to discard broth from suspected livestock with RVF, believed to harbour the virus, was a joint decision.

## Adaptive coping strategies to RVF and decision making

The participants in this study (FGDs and KIIs) reported various strategies to cope with RVF outbreaks and protect their primary livelihoods and livestock. These strategies included selling livestock, starting new businesses, accessing healthcare services, and using traditional herbal medicines. Men in all FGDs reported that selling livestock was an important strategy to adapt to the effects of RVF during an outbreak. Male further reported that livestock sales provided them with an opportunity to generate income and ensure survival amidst the crisis. In addition, male participants and all KII participants revealed that although livestock sales provided income opportunities to buy foodstuff for households, the livestock markets were limited during outbreaks due to quarantine measures implemented to control the spread of the disease. However, despite these limitations, men's self-reported reasons for this were fear of failing to provide for their families and fear of losing all their livestock. In male-headed households, participants in all FGDs mentioned that men traditionally make the final decision regarding livestock sales. This decision-making authority is rooted in both customary gender roles and the desire to maintain household harmony. All FGD participants revealed that men discussed their intentions to sell livestock with their spouses, but the final decision rested with them:
*"For us not to have conflicts in our family, we sit, and I inform the women which livestock is due*

*for selling and how many I would want to sell, but I (man) make the final decision" (Men-MHH FDG02).*

In contrast, participants in all FGDs highlighted that woman from male-headed households had more influence over poultry and household domestic-related decisions, while men primarily controlled the sale of valuable livestock. Women from the FHH reported that they were primary owners and had decision-making power over the livestock they owned. This was because these women did not have men, and they were the primary caregivers. However, although all women from FHH were reported to be primary decision makers, their decision-making power was not entirely static. Factors such as the presence of older sons or proximity to other family members have been reported to influence decision-making power, leading to joint decisions. Although decision-making was reported to not be static, women retained the final say in how funds from livestock sales were utilized, emphasizing their financial decision-making authority.

In addition, women in all FGDs reported venturing into new businesses as a critical coping mechanism for pastoralist communities. These women further noted that they established small businesses, such as vegetable stalls and general merchandise shops, within their households to secure their families' survival and meet essential needs during and after the RVF outbreak. Decisions to start new businesses with women were reported to be made jointly by couples, with men providing financial resources and women taking on operational roles. This is illustrated in the following narration:

*"RVF and the effects of long droughts have taught us to start new small businesses in our household which help us survive when animals are sick or dead. The decision to begin these businesses is often agreed upon between husband and wife". (Women from MHH FGD 02).*

Participants from all FGDs and KIIs indicated that access to health facilities was a key strategy for coping with RVF outbreaks. All participants confirmed that they sought medical attention at government hospitals, private hospitals, or chemists when household members became ill during an RVF outbreak. Most male and female participants demonstrated that they preferred to seek care at government hospitals because of the availability of on-site laboratories for blood testing and comprehensive healthcare services. This finding was confirmed by KII participants, who disclosed that they encouraged the general population to seek health services from government facilities closer to them for proper diagnosis. All FGD participants reported that women and children in male-headed households often require permission from male household heads (husbands or fathers) to access medical care. This restriction stemmed from the male household head's position as the primary resource holder and decision maker within the household. As one female participant noted, "*Without a man's decision, nothing works, whatever the man says is right. So, he decides when to take the sick person to the hospital*" (Female herders FGD01). However, the men in all FGDs established exceptions to their sole decision-making authority in access to healthcare. Men noted that these exceptions were limited to emergencies or unavailability. In cases where men were not available, women were reported to make healthcare decisions, but were expected to inform their husbands after making these decisions. In addition, all men and women consented that they relied on readily available herbal medicines when their access to formal healthcare was limited. Although men and women expressed confidence in herbal medicines, healthcare officials demonstrated concerns regarding their efficacy and safety. *"We as medical officers, we discourage the use of herbals or traditional medicines to treat RVF, because we don't know how it works" (KII human health officials).*

### Transformative resilience to RVF and gendered decision making

In this study, FGD and KII participants reported three transformative resilience strategies: access to information, RVF, surveillance, and vaccination, which they relied on to build resilience to RVF, as elaborated.

Participants in all FGD and KII groups indicated that access to knowledge before, during, and after an RVF outbreak is a key strategy for effective response and prevention of RVF exposure in the community. All FGD participants, mainly men, reported that they actively shared information about RVF signs and symptoms, and prevention strategies among themselves. All women participants reported that, when they observed sick animals in their kraal, they informed their husbands(men). All male participants indicated that when they observe RVF signs in their herds or household members, they quickly share this information within their community and seek guidance from healthcare professionals, Community Disease Reporters (CDRs), Community Health Volunteers (CHVs), and fellow pastoralists. These findings are in agreement with the KII (CDRs and CHVs), confirming that pastoralist men often inform them about the suspected signs and symptoms of RVF in livestock and humans, who then later inform county veterinary/public health officers for response. In addition, all FGD men reported that when livestock showed RVF symptoms, the affected animals were isolated from healthy animals, and in some cases, healthy livestock were relocated to safer areas. Men reportedly make decisions to relocate or isolate livestock.

All FGD and KII participants revealed disease surveillance as a cornerstone strategy for building resilience against RVF outbreaks. All participants reported the presence of trained CHVs and CDRs in the communities that helped with disease surveillance in the absence of county veterinary officers. FGD in men and women illustrated the critical role of CHVs and CDRs in effectively educating the community about RVF, exposure risks, signs and symptoms of RVF, and preventive measures. The KIIs (CDRs and CHVs) reported that they actively engaged with the community to identify potential cases and promptly reported suspected cases to the Isiolo County government for early response.

Men in all groups noted that vaccination of livestock was an effective strategy for protecting livestock from RVF and preventing its transmission to humans. Pastoralist men reported that RVF vaccines boost livestock immunity and provide long-lasting protection but are not readily available. In addition, all men noted that they preferred to vaccinate their animals during the rainy season, as mosquito breeding increased following heavy rainfall, increasing the risk of RVF transmission. Importantly, men have the final decision-making authority regarding livestock vaccination. The decision stems from livestock ownership, their perceived greater knowledge about livestock care, and the practical challenges faced by women in traveling to distant vaccination centers and managing household responsibilities.

## Discussion

Our study investigated how gender relations influence resilience to Rift Valley Fever (RVF) in pastoralist communities in Isiolo County, Kenya. This study revealed gender-based differences in livestock ownership and decision-making power, which significantly influence individual resilience to RVF. The participants in this study noted that men predominantly own livestock, particularly larger animals like cows, goats, and camels, have sole decision-making power within the household. In contrast, women primarily own poultry and donkeys, and their decisions are contingent upon their husbands. Our findings are consistent with those of previous studies that highlighted women's ownership rights and limited decision-making power [1,18]. This disparity stems from the cultural norms and traditions related to inheritance rights and household headships. These variations in ownership and decision-making can impede the

resilience process, increase vulnerability, and reduce the ability to withstand RVF shocks during an outbreak. These findings corroborate those of previous studies [19,20] that identified a correlation between men's leadership roles and livestock ownership, and greater resilience against RVF in pastoralist communities. The economic implications of this disparity disproportionately affect who, owing to limited livestock holdings, have fewer resources to absorb losses, resulting in increased vulnerability. Some studies have found that women's limited ownership and decision-making authority can hinder their ability to cope with and recover from shocks, such as RVF outbreaks, as livestock are crucial for income, healthcare, and community participation [34,35]. Similarly, [22] revealed that patriarchal structures, especially in pastoralist communities, loosen women's bargaining power to RVF outbreaks and have a long-term effect on women's well-being and preparedness to mitigate the impact of RVF outbreaks and build resilience, as evidenced in the present study.

Regarding absorptive capacities, our study examined gender dynamics in the adoption of preventive measures, such as mosquito nets, repellents, and hygiene practices. Men and women use mosquito nets and repellents to reduce the risk of exposure to RVF infections, and the decision to use these measures is jointly made by spouses. The participants in this study argued the importance of equal protection for their household, healthy community environment, and economic impact. The findings of this study resonate with those of previous research on gendered health responsibilities [36–38]. Our findings show that men and women are aware of the importance of working together to protect their lives and livelihoods (livestock) and promote economic stability if RVF is well managed and controlled within their communities. This is consistent with a previous study that emphasized the importance of the joint decision-making process within households as a crucial component of risk-mitigation strategies during zoonotic disease outbreaks [39], emphasizing the collaborative nature of gendered roles in promoting health and safety within pastoralist communities. Our findings also contribute to the existing literature by highlighting the critical role of gender in health decision making and risk mitigation within communities. In addition, good hygiene practices, such as boiling milk and meat, clearing bushes, and using gloves, were deemed crucial for mitigating RVF's effects of the RVF. These practices are often collaborative efforts within households, with women taking the lead in certain areas and men providing support, demonstrating the significance of research on gendered health responsibilities in disease management and family well-being. Our findings also support the broader literature on the importance of collaborative approaches, including men's involvement in health care uptake. to improve health outcomes [39–41].

In terms of adaptive capacities, this study identifies gender inequalities in the adaptation of pastoralist communities to RVF. The male participants noted that they sell livestock either alive or slaughtered to generate income during RVF outbreaks to sustain their families. The decision regarding livestock sales solely depends on the men, but they face the challenge of limited markets due to quarantine measures. This finding resonates with a study on pastoralists' perceptions of the impact of the RVF in Kenya [5]. Quarantine measures directly affected them because of their dependence on livestock as their main source of food and livelihood. The partial closure of livestock markets due to the RVF outbreak without government intervention results in limited access to food, income, and health which are key factors in mitigating the impact of RVF in pastoralist communities and in the long run, may increase their vulnerability exposure to RVF. While men dominate decision making in livestock sales, women in female-headed households have significant decision-making power. This finding aligns with previous studies on gendered intra-household decision making, where women from female-headed households have greater autonomy in household decision making than women from male-headed households [1,42]. Apart from livestock sales, women engage in alternative

livelihoods, such as starting small businesses, to cope with the economic impacts of the RVF. Through business diversification, women can enhance their families' well-being, particularly by providing basic needs, such as food. Women's ability to manage their businesses empowers them and strengthens their resilience against challenges, such as RVF outbreaks. This finding aligns with those of some studies on building resilience to climate change in Kakamega-Kenya, which found that women engage in different activities to diversify their resilience to climate change [29,43]. However, we found that women's ability to start businesses and diversify their income sources was constrained by their access to financial resources. This is because women rely on their husbands for financial support, and the decision to provide funds ultimately rests with men. This limited access to resources can hinder women's capacity to build resilience against challenges, such as RVF outbreaks. While access to healthcare services is a key coping strategy, men often make decisions about seeking medical care from household members, although women may influence these decisions in certain situations in the absence of men. This collaborative approach aligns with previous studies on autonomy and couples' joint decision making in healthcare [44,45]. However, cultural norms and limited access to resources can hinder women's access to healthcare services during RVF outbreaks. These challenges to healthcare access are similar to those identified in a study of women in pastoral societies [46].

A cross-sectional study design was used to examine how ownership and household decisions influence men's and women's resilience capacities. The researcher did not collect data during or immediately after the RVF outbreak to interview pastoralist men and women on the adaptation strategies they employed to respond to RVF and make the final decisions regarding the adapted strategies. All the data collected were drawn from previously remembered experiences. It can be argued that the information gathered is helpful because it enables us to comprehend how men and women respond to and reduce the adverse effects of RVF and who in the household has the final say. Moreover, this information is useful in understanding how livestock ownership influences the resilience capacities of pastoralist men and women.

This study found that transformative capacities are also influenced by gender, where both men and women actively share information about RVF within their communities, contributing to disease awareness and response. Knowledge sharing among men and women was common, especially when men and women shared information on crucial adaptation strategies with their spouses, fellow women, or men. The most valuable resource upon which pastoralists rely to receive information about RVF is the Community Health Volunteers (CHVs), and Community Disease Reporters (CDRs). In Isiolo County, nearly every village has CDRs and CHVs that play crucial roles in disease surveillance and community engagement, fostering a transformative approach to disease management. This collective knowledge dissemination approach aligns with the findings of a previous study on health service uptake among nomadic pastoralist African populations. This study emphasizes the pivotal role of community-based knowledge exchange in enhancing disease awareness and response within pastoralist communities [44]. These community-based groups provide social support and networks to men and women, including early warning, surveillance, information on signs and symptoms, management strategies, and early reporting of suspected RVF cases to the Isiolo County Government Department of Veterinary and Public Health before, during, and after RVF suspected outbreaks. Through these community-based efforts, men and women have been able to identify suspected RVF cases in humans and livestock and take precautions to protect themselves, despite other challenges. This is in line with the previous literature on the importance of social support for disease management, which ensures equitable access to information for all community members, enhances community preparedness, mitigates the impact of RVF outbreaks, and builds resilience. In addition to information dissemination, livestock vaccination has been considered an effective strategy for preventing RVF in humans and livestock. Livestock

vaccination protects the community, even during rainy seasons when mosquitoes breed. This finding aligns with [25,47] who highlight the importance of livestock vaccination. However, while vaccination strategies have the potential to protect livestock and prevent RVF transmission to humans, this study identifies significant barriers, such as unequal decision-making dynamics, where men typically make decisions about livestock vaccination due to their ownership and perceived expertise, and limited access to RVF vaccines. Our findings resonate with those of [47], emphasizing the need for gender-equitable decision-making processes in livestock management and disease prevention strategies within pastoralist communities.

## Conclusion

In conclusion, this study revealed a complex interplay of gender relations that influences resilience to RVF in pastoralist communities. These findings suggest that addressing gender inequalities in livestock ownership, decision making, and resource access is essential for enhancing community resilience to future outbreaks. Absorptive strategies, such as mosquito nets and implementation of hygiene practices, are subject to gendered decision-making within households. Adaptive strategies, including livestock sales and the establishment of new businesses, help mitigate the immediate impacts. Transformative strategies such as information sharing, disease surveillance, and vaccination foster long-term protection. Collaboration between men and women is crucial to the success of these strategies. This research highlights the importance of integrating gender-sensitive measures into future interventions to strengthen pastoralist livelihoods. This includes gender-inclusive training, implementation of support programs, and development of robust community-based disease surveillance and vaccination initiatives. Future research could explore the economic impact of women's income generation, generational shifts in decision making, and the efficacy of traditional herbal medicines alongside conventional treatments for RVF.

## Supporting information

**S1 Dataset. Transcripts.**
(ZIP)

## Acknowledgments

The authors extend their gratitude to the project team, community leaders, and members of the Merti and Garbatulla sub-counties, Isiolo County Veterinary Department, Public Health Department, County Hospital, and Gender Department. Special appreciation goes to the enumerators who worked tirelessly to ensure the smooth running of the data collection.

## Author Contributions

**Conceptualization:** Irene N. Mutambo, Bernard Bett, Salome A. Bukachi.

**Data curation:** Irene N. Mutambo.

**Formal analysis:** Irene N. Mutambo, Bernard Bett, Salome A. Bukachi.

**Funding acquisition:** Salome A. Bukachi.

**Investigation:** Irene N. Mutambo.

**Methodology:** Irene N. Mutambo, Bernard Bett, Salome A. Bukachi.

**Project administration:** Salome A. Bukachi.

**Resources:** Bernard Bett, Salome A. Bukachi.

**Supervision:** Bernard Bett, Salome A. Bukachi.

**Validation:** Bernard Bett, Salome A. Bukachi.

**Visualization:** Irene N. Mutambo.

**Writing – original draft:** Irene N. Mutambo.

**Writing – review & editing:** Irene N. Mutambo, Bernard Bett, Salome A. Bukachi.

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
