## [Decision Letter · Decision Letter 0]

18 Jun 2024

PONE-D-24-03443Without a Man's Decision, Nothing Works: Building Resilience to Rift Valley fever in Pastoralist Communities in Isiolo KenyaPLOS ONE

Dear Dr. mutambo,

Thank you for submitting your manuscript to PLOS ONE. After careful consideration, we feel that it has merit but does not fully meet PLOS ONE’s publication criteria as it currently stands. Therefore, we invite you to submit a revised version of the manuscript that addresses the points raised during the review process.

We look forward to receiving your revised manuscript.

Kind regards,

Nasrin Akter, MPH

Guest Editor

PLOS ONE

Journal Requirements:

2. Thank you for stating the following financial disclosure: "This work was supported by the One Health Research, Education and Outreach Centre in Africa funded by The Federal Ministry of Economic Cooperation and Development (BMZ) and led by International Livestock Research Institute, Nairobi, Kenya"

Reviewers' comments:

Reviewer's Responses to Questions

**Comments to the Author**

1. Is the manuscript technically sound, and do the data support the conclusions?

Reviewer #1: Partly

Reviewer #2: Yes

Reviewer #3: Partly

2. Has the statistical analysis been performed appropriately and rigorously? 

Reviewer #1: I Don't Know

Reviewer #2: Yes

Reviewer #3: Yes

3. Have the authors made all data underlying the findings in their manuscript fully available?

Reviewer #1: Yes

Reviewer #2: Yes

Reviewer #3: Yes

4. Is the manuscript presented in an intelligible fashion and written in standard English?

Reviewer #1: No

Reviewer #2: Yes

Reviewer #3: Yes

5. Review Comments to the Author

Reviewer #1: This study is a qualitative study that aimed to understand the gendered difference in pastoral population resilience to Rift Valley fever outbreaks in a high-risk area of Kenya that has experienced numerous outbreaks since 1998. The authors use a resilience capacities framework to evaluate their findings. In total, 16 focus group discussions were carried out and 13 key informant interviews which requires significant organization and implementation. However, in the current way the manuscript is written, it is difficult to understand the results of the study and background that generated the research questions. Most notably, the results are mixed in with background and discussion to the extent I cannot understand what the study found. The introduction is difficult to follow. A good start to reshaping this intro would be to remove all mentions of the study methods and attempt to make the sentences more direct. Each paragraph should serve the purpose of delivery a key piece of information relevant to understanding how you formed your research questions. In the results, I am unsure if I am reading results of the thematic analysis or misplaced introduction. The results text also includes significant interpretation of the findings, which such be in the discussion. An important clarification is needed on what outbreak the participants are being asked to refer to. The statement that the most recent outbreak in Isiolo was in 2007 is factually incorrect.

This feels like an important research question, but with the disorganization of the sections of manuscript as is, it is unclear what the key study findings are and what the author has interpreted. There are no mentions of data saturation or how many groups contributed to the different results mentioned. I highly encourage the author to resubmit the manuscript after making these changes as it seems like this is important work that could be a valuable contribution to RVF literature. Unfortunately, there are too many mistakes in the structure of the paper to give the study findings a fair evaluation. I hope the following comments are helpful in developing the next edition.

General comments

Do not use in text citations in the place of the authors name you are mentioning.

All figures need a title and should be labeled Figure 1, Table 1, ect. The figure should come immediately after the paragraph it is mentioned in

Fever should not be capitalized in “Rift Valley fever”

Title:

If the first part of the title is a participant quote, add quotation marks.

Abstract:

Line 11: Begin the abstract with a brief introduction of RVF and/or a summary of what is known about gendered differences in pastoral resilience. Move “this article presents…” to the end of the abstract

Line 18: “We established that pastoralists use different approaches to reduce the effects of RVF on their livelihoods and ensure continuity” Pastoralist use different approached than who? Or men and women use different approached? This sentence is vague, and it would be best to re-word. Has the study established this or was it already known?

Line 31: Change can be to “is” and you may also want to note that RVF can be directly transmitted to humans from animals

Line 33: Best to avoid refereeing to RVFV as "it". You could change “and it usually” to “impacting”

Line 35: Instead of (8), write the authors name if this is an article “Name and others.” However, this is basic RVF knowledge, and you can just write the sentence and the citation at the end

Line 39: Remove “However”

Line 43: I disagree that the most recent outbreak was 2007. See 2018, 2020, now…?

Line 44: I also disagree with this statement that the 2007 outbreak mainly affected the NE.

Line 46: Remove headings from introduction.

Line 47: Indent new paragraph

Line 49: Remove “are mainly” if they are primary. This is repetitive

Line 52: I am not an expert in women decision making, but I believe it is known that women are involved heavily in decisions around child health and nutrition. You may mention this to highlight that women make decisions beyond household chores

Line 54: Adaptation to what?

Line 56: start new sentence with “Hence,

Line 60: Remove mentions of your study’s method from the introduction. I would start this paragraph at line 61 rather than with the definition of resilience

Line 61: Add the word “framework,” “theory” or something similar after “social-ecological resilience”

Line 64: New sentence “For example,”

Line 66: Add “of between examples and abortive and replace “might have” with “include” since you are giving an example.

Line 73: It becomes repetitive to read “examples” you can shorten this by giving a simple example in the preceding sentence by using “such as”

Line 75: Again, this is the introduction, remove all mentions of your study methods.

Line 78: This section needs to be in the same paragraph as when you first mention gender and womens’ roles. It interrupts the flow here. There is a typo here also

Line 85: This paragraph also does not seem to fit here. Perhaps move these mentions of challenges above the part where you talk about building resilience.

Line 97: Be more specific on the type of RVF studies because most RVF studies overall are not focused on KAP, they are epidemiological studies. This paragraph could be deleted entirely as it does not add to the background of your study. It is fine to state that your study is a good idea from line 101 onward but make it shorter and focus on the potential impact of having such data. Then, follow with the objective of this study in one line.

Line 106: Your research questions do not belong in the introduction. End the introduction with the objective/intent of your research.

Methods

Line 113: You could start the methods with an overview of your study moving everything from the introduction

Line 120: Confirm if you reviewed the consent form orally with those participants that were unable to read and the languages consent forms were delivered in.

Line 122: If the key information here is that your study area is pastoralist. It would be good to give information about livestock ownership, what proportion of households are classified as pastoral/semi-pastoral. I believe this is available from the 2019 census

Line 124: This is background information you have already provided in the intro. Delete

127: Deployed?

121: Data collection was “carried out”

131: “a vignette” You could combine this sentence with line 132 to improve flow

137: This figure is presented without a title and does not follow logically. It interrupts this description of the vignette you are trying to give. Figures should be placed immediately after the paragraph they are mentioned in. This looks like it belongs with study site. Furthermore, the scale you have presented the landcover at is not easy to interpret. It would be best to choose the most prominent landcover types. I don’t see any open water on the map, so it should not be in the legend.

140: Were males and females questioned in the same space? Were any measures in place to ensure they were revealing their cards at the same time and that bias was indeed reduced?

146: Table 1 is also misplaced. It should be placed after this paragraph where you mention it in the text.

154: “clean them up” is too informal. State the modifications that were made

Line 161: I have just realized that you have many results in the methods section. Remove all mentions of results in the methods, for example table 1.

Results

163: Did all study participants own all these animals? Include numbers and proportions if this is a finding.

166/167: Was this a finding from your data collection or is this cultural norm something that is known? If so, it belongs in the introduction where you talk about gendered differences.

170: It would be best to give numbers and percents here as this seems highly relevant to your findings.

171: The part where you said “this is attributed” is an interpretation of results. This belongs in the discussion

172: Describing how women typically acquire livestock is background information. This belongs in the introduction unless your study identified this as a theme. If it is the later, it would be best to put this under a heading that you are leading into the results of the thematic analysis. The same goes from like 175 onward. I am unsure if this is a result of your study or misplaced introduction/interpretation of your results.

185: This is another example of introduction in the results. It makes it very difficult to follow what this study has found.

190: Of what RVF outbreak? If you mean outbreaks in general, add an s to outbreak. It would also be impactful to describe what outbreak they are refering to. Above you mention that your interpretation is the last outbreak was in 2007, if this is the one the participant is referring to, this is quite a long recall time. Depending on the above answer, you may consider including recall bias in your limitations section of the discussion.

192: Did the participants say this or is it your interpretation? If a participant said this, you can re-word “In X/X groups, participants noted that…”

206: Here is another example where it is hard to understand if this is a result. Results are generally in the past tense. In line 209, you then say the participants reported something, so it makes the reader understand that everything prior to that was misplaced background information.

219: Define what the soup is from slaughter

225: Another example of background text. I will stop pointing this out from this line onwards as I can’t interpret the findings of the study.

Discussion

310 and 325: Put the authors name instead of an in-text citation

398: Limitations should be a paragraph in the discussion rather than a separate section. I’d suggest the second to the last paragraph.

418: All references need to have their format checked. The first one is all capital letters.

297: Did the participants say this or this this a general statement on RVF vaccination?

Reviewer #2: 1. In line number 99-100 indicating "there are aslo a few studies......" need reference.

2. Please mention future implication after the objective of the study.

3. Provide ethical statement at the last portion of the study

4. Add more reference in the discussion section

5. Add limitation in the discussion section

6. Please check overall English language

Reviewer #3: Need to generate latest data, focus on study design again and sample size determination. Conclusion should be result oriented with focus one future implication. Discusssion part can be short based on key findings.

6. PLOS authors have the option to publish the peer review history of their article (what does this mean?). If published, this will include your full peer review and any attached files.

Reviewer #1: No

Reviewer #2: **Yes: **NUSRAT HOSSAIN SHEBA

Reviewer #3: No

---

## [Author Response · Author response to Decision Letter 0]

25 Jul 2024

Rebuttal Letter

Dear Editor,

Thank you for the opportunity to revise and resubmit our manuscript, "”Without a Man’s Decision, Nothing Works”: Building Resilience to Rift Valley fever in Pastoralist Communities in Isiolo Kenya" [PONE-D-24-03443]. We appreciate the time and valuable feedback provided by the reviewers and editorial team. We have carefully addressed each concern, and we believe these revisions have significantly strengthened the manuscript.

Enclosed please find the following:

• Response to Reviewers: A point-by-point response to each comment, outlining the specific changes made in the manuscript.

We are confident that our revised manuscript now meets the high standards of PLOS One. We are grateful for your consideration and look forward to your decision.

Sincerely,

Nabwire Irene Mutambo 

General comments

Comment: Do not use in text citations in the place of the authors name you are mentioning.

Response: We have adopted the PLOS ONE referencing style

Comment: All figures need a title and should be labeled Figure 1, Table 1, ect. The figure should come immediately after the paragraph it is mentioned in

Response: All the figures/titles have been given titles and moved to the appropriate places

General comments

Do not use in text citations in the place of the authors name you are mentioning.

All figures need a title and should be labeled Figure 1, Table 1, ect. The figure should come immediately after the paragraph it is mentioned in

Comment: Fever should not be capitalized in “Rift Valley fever”

Response: All the capital F has been removed from the manuscript

Title:

Comment: If the first part of the title is a participant quote, add quotation marks.

Response: Yes, the quote was picked from the participant’s response. The quotation has been added (though the journal system doesn’t allow quotes while submitting and that is how I submitted the title without the quotes with the advice from the PLOS ONE team)

Abstract:

Comment: Line 11: Begin the abstract with a brief introduction of RVF and/or a summary of what is known about gendered differences in pastoral resilience. Move “this article presents…” to the end of the abstract

Response: The abstract now includes an introduction to RVF and highlights key gendered differences in resilience, offering a concise overview of our findings

Comment: Line 18: “We established that pastoralists use different approaches to reduce the effects of RVF on their livelihoods and ensure continuity” Pastoralist use different approached than who? Or men and women use different approached? This sentence is vague, and it would be best to re-word. Has the study established this or was it already known?

Response: This sentence has been revised to specify the different approaches used by men and women pastoralists in coping with RVF, enhancing clarity. This is a study finding

Comment: Line 31: Change can be to “is” and you may also want to note that RVF can be directly transmitted to humans from animals

Response: The adjective “is” has been adopted as requested and the statement has been revised

Comment: Line 33: Best to avoid refereeing to RVFV as "it". You could change “and it usually” to “impacting”

Response: Referring has been removed from the sentence

Comment: Line 35: Instead of (8), write the authors name if this is an article “Name and others.” However, this is basic RVF knowledge, and you can just write the sentence and the citation at the end

Response: This is a referencing/citation style for PLOS ONE. The citation has been put at the end of the sentence

Comment: Line 39: Remove “However”

“However,” has been removed

Comment: Line 43: I disagree that the most recent outbreak was 2007. See 2018, 2020, now…?

Response: We have updated this section with details of subsequent outbreaks in 2018, 2020, and 2024, providing a more complete historical context.

Comment: Line 44: I also disagree with this statement that the 2007 outbreak mainly affected the NE.

Response: This statement has been reworded and the 2020 outbreak that affected Isiolo county has been captured

Comment: Line 46: Remove headings from introduction.

Response: The heading has been removed

Line 47: Indent new paragraph

Response: The new paragraph has been indented after the heading has been removed

Comment: Line 49: Remove “are mainly” if they are primary. This is repetitive

Response: “Mainly” has been removed”

Comment: Line 52: I am not an expert in women decision making, but I believe it is known that women are involved heavily in decisions around child health and nutrition. You may mention this to highlight that women make decisions beyond household chores

Response: Children’s health and nutrition has been introduced as suggested by the reviewer

Comment: Line 54: Adaptation to what?

Response: The sentence has been revised for clarity

Comment: Line 56: start new sentence with “Hence,

Response: This whole sentence has been revised

Comment: Line 60: Remove mentions of your study’s method from the introduction. I would start this paragraph at line 61 rather than with the definition of resilience

Response: The method has been removed and the definition of resilience.

Comment: Line 61: Add the word “framework,” “theory” or something similar after “social-ecological resilience”

Response: The word “theory” has been introduced after “social-ecological resilience”

Comment: Line 64: New sentence “For example,”

Response: A new sentence has been introduced with “For example”

Comment: Line 66: Add “of between examples and abortive and replace “might have” with “include” since you are giving an example.

Response: This sentence has been revised to address all the grammatical errors

Comment: Line 73: It becomes repetitive to read “examples” you can shorten this by giving a simple example in the preceding sentence by using “such as”

Response: The sentence has been shortened and conjunctions (for instance, such as” have been used to replace “example” 

Comment: Line 75: Again, this is the introduction, remove all mentions of your study methods.

Response: Methods have been removed from the introduction

Comment: Line 78: This section needs to be in the same paragraph as when you first mention gender and womens’ roles. It interrupts the flow here. There is a typo here also

Response: This paragraph has been moved and placed after the indented paragraph. In addition, this paragraph has been revised to enhance clarity

Comment: Line 85: This paragraph also does not seem to fit here. Perhaps move these mentions of challenges above the part where you talk about building resilience.

Response: This paragraph has been moved to the third after the indented paragraph

Comment: Line 97: Be more specific on the type of RVF studies because most RVF studies overall are not focused on KAP, they are epidemiological studies. This paragraph could be deleted entirely as it does not add to the background of your study. It is fine to state that your study is a good idea from line 101 onward but make it shorter and focus on the potential impact of having such data. Then, follow with the objective of this study in one line.

Response: The paragraph has been deleted as suggested by the reviewer

Comment: Line 106: Your research questions do not belong in the introduction. End the introduction with the objective/intent of your research.

Response: The research questions have been deleted from the manuscript and the study objective has been put at the end of the introduction

Methods

Comment: Line 113: You could start the methods with an overview of your study moving everything from the introduction

Response: An overview of the study has been provided

Comment: Line 120: Confirm if you reviewed the consent form orally with those participants that were unable to read and the languages consent forms were delivered in.

Response: Yes, the consent forms were reviewed orally in the local language with all the participants regardless of whether an individual was able/unable to read. The sentence has been revised for clarity

Comment: Line 122: If the key information here is that your study area is pastoralist. It would be good to give information about livestock ownership, what proportion of households are classified as pastoral/semi-pastoral. I believe this is available from the 2019 census

Response: This information has been provided

Comment: Line 124: This is background information you have already provided in the intro. Delete

Response: This information has been deleted

Comment: 127: Deployed?

Response: This section has been revised to enhance clarity

Comment: Do not use in text citations in the place of the authors name you are mentioning.

Response: We have adopted the PLOS ONE referencing style

Comment: All figures need a title and should be labeled Figure 1, Table 1, ect. The figure should come immediately after the paragraph it is mentioned in

Response: All the figures/titles have been given titles and moved to the appropriate places

Comment: 121: Data collection was “carried out”

Response: The section on the “study area” was reworked and information was added for a good flow of information

Comment: 131: “a vignette” You could combine this sentence with line 132 to improve flow

Response: Line 131 and 132 have been combined to improve the structural flow of the argument

Comment: 137: This figure is presented without a title and does not follow logically. It interrupts this description of the vignette you are trying to give. Figures should be placed immediately after the paragraph they are mentioned in. This looks like it belongs with study site. Furthermore, the scale you have presented the landcover at is not easy to interpret. It would be best to choose the most prominent landcover types. I don’t see any open water on the map, so it should not be in the legend.

Response: A title has been provided to the figure and the figure has been moved next to the study site. The map plus the scale have been revised

Comment: 140: Were males and females questioned in the same space? Were any measures in place to ensure they were revealing their cards at the same time and that bias was indeed reduced?

Response: We had separate groups for men and women and this has been well described in the current manuscript

Comment: 146: Table 1 is also misplaced. It should be placed after this paragraph where you mention it in the text.

Response: The table has been deleted

Comment: 154: “clean them up” is too informal. State the modifications that were made

Response: The use of formal language in the sentence has been adapted.

Comment: Line 161: I have just realized that you have many results in the methods section. Remove all mentions of results in the methods, for example table 1.

Response: All the results have been removed from the results section and Table 1 has been deleted because the information in the table is elaborated in the naration

Results

Comment: 163: Did all study participants own all these animals? Include numbers and proportions if this is a finding.

Response: Yes, all the study participants reported ownership of various livestock. This paragraph has been revised to specify the different livestock ownership by men and women and the proportion has been provided, enhancing clarity 

Comment: 166/167: Was this a finding from your data collection or is this cultural norm something that is known? If so, it belongs in the introduction where you talk about gendered differences.

Response: Yes, the finding was from data collection. This has been reworked for clarity

Comment: 170: It would be best to give numbers and percents here as this seems highly relevant to your findings.

Response: Percentages have been provided

Comment: 171: The part where you said “this is attributed” is an interpretation of results. This belongs in the discussion

Response: This statement has been removed and all interpretation statements have been removed from the results section

Comment: 172: Describing how women typically acquire livestock is background information. This belongs in the introduction unless your study identified this as a theme. If it is the later, it would be best to put this under a heading that you are leading into the results of the thematic analysis. The same goes from like 175 onward. I am unsure if this is a result of your study or misplaced introduction/interpretation of your results.

Response: This statement highlights the reasons why women own what they own but not an introduction. The sentence has been reworded to reflect the study findings

Comment: 185: This is another example of introduction in the results. It makes it very difficult to follow what this study has found.

Response: The information was deleted

Coment: 190: Of what RVF outbreak? If you mean outbreaks in general, add an s to outbreak. It would also be impactful to describe what outbreak they are refering to. Above you mention that your interpretation is the last outbreak was in 2007, if this is the one the participant is referring to, this is quite a long recall time. Depending on the above answer, you may consider including recall bias in your limitations section of the discussion.

Response: The sentence was revised and “an” was added. The introduction was revised to capture the most recent RVF outbreaks of 2018, 2020, and 2024. However, for this study, the participants referred to the 2020 RVF outbreak that affected their community. Recall bais was one of the study limitations

Comment: 192: Did the participants say this or is it your interpretation? If a participant said this, you can re-word “In X/X groups, participants noted that…”

Response: The sentences have been revised to indicate which study participants (FGD/KII) reported the finding

Comment: 206: Here is another example where it is hard to understand if this is a result. Results are generally in the past tense. In line 209, you then say the participants reported something, so it makes the reader understand that everything prior to that was misplaced background information.

Response: The sentence has been reworded to capture the participant's responses/study findings

Comment: 219: Define what the soup is from slaughter

Response: The definition of soup has been provided and the sentence has been revised

Comment: 225: Another example of background text. I will stop pointing this out from this line onwards as I can’t interpret the findings of the study.

Response: The paragraph has been revised to reflect the study findings

Comment: 297 Did the participants say this or this this a general statement on RVF vaccination?

Response: Yes, the study participants reported vaccination as a coping strategy. The sentence has been revised

Discussion

Comment: 310 and 325: Put the authors name instead of an in-text citation

Response: The in-text citation is the preference of the journal

Comment 398: Limitations should be a paragraph in the discussion rather than a separate section. I’d suggest the second to the last paragraph.

Response: The limitations of the study have been moved to the discussion section

Comment: 418: All references need to have their format checked. The first one is all capital letters.

Response: All references have been checked, and formatted

Reviewer #2: 

Comment: 1. In line number 99-100 indicating "there are aslo a few studies......" need reference.

Response: Reviewer#1 suggested that line numbers 97-100 should be removed. Therefore, the information that needed a reference was removed

Comment: 2. Please mention future implication after the objective of the study.

Response: The future implication of the study has been provided and placed after the study objective

Comment: 3. Provide ethical statement at the last portion of the study

Response: The ethical statement has been moved from the method section to the last portion of the manuscript

Comment:4. Add more reference in the discussion section

Response: More references have been added throughout the introduction and discussion

Reference numbers: 12, 14,15,16,32,34,36,37,41,42,43,45,46,47,57, and 59

Comment: 5. Add limitation in the discussion section

Response: The limitations of the study were moved to the discussion section

Comment: 6. Please check overall English language

Respo

---

## [Decision Letter · Decision Letter 1]

28 Aug 2024

PONE-D-24-03443R1Without a Man's Decision, Nothing Works: Building Resilience to Rift Valley fever in Pastoralist Communities in Isiolo KenyaPLOS ONE

Dear Dr. mutambo,

Thank you for submitting your manuscript to PLOS ONE. After careful consideration, we feel that it has merit but does not fully meet PLOS ONE’s publication criteria as it currently stands. Therefore, we invite you to submit a revised version of the manuscript that addresses the points raised during the review process.

**Please revise and update your manuscript according to the feedback of the reviewers to make it more scholarly and eligible for publication (see Reviewer #1 & Reviewer #3's feedback). Please **
**improve language throughout the manuscript to be less colloquial. **

We look forward to receiving your revised manuscript.

Kind regards,

Nasrin Akter, MPH

Guest Editor

PLOS ONE

Reviewers' comments:

Reviewer's Responses to Questions

**Comments to the Author**

1. If the authors have adequately addressed your comments raised in a previous round of review and you feel that this manuscript is now acceptable for publication, you may indicate that here to bypass the “Comments to the Author” section, enter your conflict of interest statement in the “Confidential to Editor” section, and submit your "Accept" recommendation.

Reviewer #1: All comments have been addressed

Reviewer #2: All comments have been addressed

Reviewer #3: (No Response)

2. Is the manuscript technically sound, and do the data support the conclusions?

Reviewer #1: Yes

Reviewer #2: Yes

Reviewer #3: No

3. Has the statistical analysis been performed appropriately and rigorously? 

Reviewer #1: Yes

Reviewer #2: Yes

Reviewer #3: No

4. Have the authors made all data underlying the findings in their manuscript fully available?

Reviewer #1: Yes

Reviewer #2: Yes

Reviewer #3: No

5. Is the manuscript presented in an intelligible fashion and written in standard English?

Reviewer #1: Yes

Reviewer #2: Yes

Reviewer #3: No

6. Review Comments to the Author

Reviewer #1: Summary

This is a revision of an original submission. The author has made significant improvement on the organization and language within the manuscript. I commend their efforts and am glad to accept this with minor revisions. A few suggestions on the minor edits for each section are highlighted below.

Title

Some inconsistency with capital letters. Would only capitalize: Without, Building, Rift Valley, Isiolo, and Kenya

Abstract

Much stronger abstract, well done!

Introduction

Line 32: remove the word “directly”

Line57: What are do’s and don’ts? Is this a theory? You could consider starting this sentence with “Based on the theory of x, ___” or take this part out completely. It looks like you are simply trying to make the point that when women have control of resources, they can cope better and negotiate…?

Line 74: End sentence after “shocks” to remove run on sentence

Line 80: End sentence after “occur” to remove run on sentence.

Line 92: From “This study’s..” until line 98: This should be in the discussion rather than introduction.

Materials and methods

Line 101: end sentence after Kenya. The explanation of RVF does not need to be repeated in methods

Line 127: Minor.. but the length of groups is a result, consider moving down

Results

Line 168: If men owned 65% of the livestock, does this mean women owned 35% of all recorded livestock? Adding numerators and denominators can help with this interpretation, but it would nonetheless be nice to clarify. If men owned 40% of cows, did women own 60% of the cows, or did 40% of the men in the study own cows?

Line 203: Believe we are missing the word net here…looks like men purchase mosquitos

Line 229: Change men to male

Line 288: Comma after RVF

Discussion and conclusion look okay.

Reviewer #2: All the points have been addressed and I think it can be published after doing some language checking by an expert.

Reviewer #3: No correction at all, not addressed anything at all, simply ignored. They need to be make it clear with major revision.

7. PLOS authors have the option to publish the peer review history of their article (what does this mean?). If published, this will include your full peer review and any attached files.

Reviewer #1: No

Reviewer #2: **Yes: **Nusrat Hossain Sheba

Reviewer #3: No

---

## [Author Response · Author response to Decision Letter 1]

4 Oct 2024

Title

Comment: Some inconsistency with capital letters. Would only capitalize: Without, Building, Rift Valley, Isiolo, and Kenya

Response: The suggestion to capitalize Without, Building, Rift Valley, Isiolo, and Kenya has been adapted and the rest of the wording in the title bears small letters

Introduction

Comment: Line 32: remove the word “directly”

Response: The word directly has been removed

Comment: Line57: What are do’s and don’ts? Is this a theory? You could consider starting this sentence with “Based on the theory of x, ___” or take this part out completely. It looks like you are simply trying to make the point that when women have control of resources, they can cope better and negotiate…?

Response: The sentence has been revised and do’s and do not’s have been removed

Comment: Line 74: End sentence after “shocks” to remove run on sentence

Response: A full stop has been introduced to avoid run on sentence

Comment: Line 80: End sentence after “occur” to remove run on sentence.

Response: A full stop has been introduced to avoid run on sentence

Comment: Line 92: From “This study’s..” until line 98: This should be in the discussion rather than introduction.

Response: The statement has not been moved because, in the first round of comments, Reviewer #2 suggested I should “mention future implications after the study's objective” in the original manuscript (lines 99-100). That is why is introduced the implication here 

Materials and methods

Comment: Line 101: end sentence after Kenya. The explanation of RVF does not need to be repeated in methods

Response: The following statement after Kenya has been removed to

Comment: Line 127: Minor.. but the length of groups is a result, consider moving down

Response: The length of the groups has been moved to lines 132-134

Results

Comment: Line 168: If men owned 65% of the livestock, does this mean women owned 35% of all recorded livestock? Adding numerators and denominators can help with this interpretation, but it would nonetheless be nice to clarify. If men owned 40% of cows, did women own 60% of the cows, or did 40% of the men in the study own cows?

Response: Numerators and denominators have been provided to help with this interpretation and presentation

Comment: Line 203: Believe we are missing the word net here…looks like men purchase mosquitos

Response: The word net has been introduced

Comment: Line 229: Change men to male

Response: Men have been changed to male

Comment: Line 288: Comma after RVF

Response: A comma has been introduced after RVF

Reviewer #2

Comment: All the points have been addressed and I think it can be published after doing some language checking by an expert.

Response: A comprehensive language review was conducted throughout the manuscript to reduce the use of colloquial language and enhance the clarity and formality of the text.

Reviewer #3: 

Comment: No correction at all, not addressed anything at all, simply ignored. They need to be make it clear with major revision.

Response: All the comments have been considered and addressed point by point as indicated in the table and in the manuscript

Comment: Focus on study design again, and sample size determination. (Where is the sample size?)

Response: The whole section on “study design and data collection has been reworked for clarity and to strengthen the section

Comment: Need to mention qualitative study

Response: The concept of qualitative study has been introduced

Comment: Analysis transparency need to be more clear

Response: The whole section on “data management and analysis” has been revised to capture the details of the data analysis process

Comment: The discussion part can be short based on key findings

Response: The discussion has revised and summarized to capture key findings

Comment: A conclusion should be result-oriented with a focus one future implications. (Must be focus on key result findings & then recommendation in one or two separate sentence)

Response: The conclusion has been revised to reflect the study results and implications, and further research

Comment: Need to generate the latest data

Response: References 18, 19, 20, 21, 25, 32, 34,37, 48 and 50 have been removed and more recent literature has been introduced. The current manuscript obtains from references (2014 to 2024) except 33 and 34. The reason why these two have been maintained, we could not find a suitable reference to replace them.

---

## [Decision Letter · Decision Letter 2]

13 Nov 2024

PONE-D-24-03443R2Without a Man's Decision, Nothing Works: Building Resilience to Rift Valley fever in Pastoralist Communities in Isiolo KenyaPLOS ONE

Dear Dr. mutambo,

Thank you for submitting your manuscript to PLOS ONE. After careful consideration, we feel that it has merit but does not fully meet PLOS ONE’s publication criteria as it currently stands. Therefore, we invite you to submit a revised version of the manuscript that addresses the points raised during the review process.

We look forward to receiving your revised manuscript.

Kind regards,

Nasrin Akter, MPH

Guest Editor

PLOS ONE

Journal Requirements:

Reviewers' comments:

Reviewer's Responses to Questions

**Comments to the Author**

1. If the authors have adequately addressed your comments raised in a previous round of review and you feel that this manuscript is now acceptable for publication, you may indicate that here to bypass the “Comments to the Author” section, enter your conflict of interest statement in the “Confidential to Editor” section, and submit your "Accept" recommendation.

Reviewer #1: All comments have been addressed

Reviewer #3: All comments have been addressed

2. Is the manuscript technically sound, and do the data support the conclusions?

Reviewer #1: Yes

Reviewer #3: Yes

3. Has the statistical analysis been performed appropriately and rigorously? 

Reviewer #1: Yes

Reviewer #3: Yes

4. Have the authors made all data underlying the findings in their manuscript fully available?

Reviewer #1: Yes

Reviewer #3: Yes

5. Is the manuscript presented in an intelligible fashion and written in standard English?

Reviewer #1: Yes

Reviewer #3: Yes

6. Review Comments to the Author

Reviewer #1: This manuscript was difficult to review with the track changes in place as there are so many sections that have been moved or deleted. Nonetheless, I think I’ve been able to review it in enough detail. I see few major mistakes, but a few details throughout require looking at. Some of these are editorial and some require author input. The comment I made about in text citations with only a number in parentheses replacing the name of the study or authors name requires input from the editor. My comment from the first revision said that the authors had adopted PLOS One ref style, but I have never seen this before in any PLOS One papers.

Spell out numbers less than ten. 4=four

Line 138: Typo involving

Line 176: Typo between

Line 290: These statements still do not make sense. I believe you mean, 95% of men owned cows, 70% of men owned goats, ect? Why not write it simply like this? If the proportion is of men (not cows) then the % associated belongs next to men, not cows.

Line 468: This sentence fades away with no end?

Line 480: A citation cannot replace words. You can say, “A study in x reported___” but not just putting the citation.

Line 552: Resonates how?

Line 590: A switch to third person point of view “The researcher” Needs to be consistent throughout 1st person “We”

Line 639: Another example of these in-text citations that do not make sense.

Line 644: Another example of in-text citation

Reviewer #3: (No Response)

7. PLOS authors have the option to publish the peer review history of their article (what does this mean?). If published, this will include your full peer review and any attached files.

Reviewer #1: No

Reviewer #3: No

---

## [Author Response · Author response to Decision Letter 2]

18 Nov 2024

Methods section 

Comment: Spell out numbers less than ten. 4=four

Response: All the number less than ten have been spelt out

Comment: Line 138Typo involving

Response: The typo has been resolved

Comment: Line 176 Typo between

Response: The typo has been resolved

Results 

Comment: Line 290 These statements still do not make sense. I believe you mean, 95% of men owned cows, 70% of men owned goats, etc.? Why not write it simply like this? If the proportion is of men (not cows) then the % associated belongs next to men, not cows.

Response: The statement has been revised to improve result clarity

Discussion 

Comment: Line 468This sentence fades away with no end?

Response: The sentence has been revised to be complete

Comment: Line 480 A citation cannot replace words. You can say, “A study in x reported___” but not just putting the citation.

Response: The sentence has been revised. The in-text citation has been removed and placed at the end of the sentence.

Comment: Line 552 Resonates how?

Response: How the results resonate with our finding has been provided

Comment: Line 590 A switch to third person point of view “The researcher” Needs to be consistent throughout 1st person “We”

Response: A switch to third persons’ view have remove and replaced with “we” as suggested by the reviewer

Comment: Line 639 Another example of these in-text citations that do not make sense.

Response: The sentence has been revised. The in-text citation has been removed and placed at the end of the sentence.

Comment: Line 644 Another example of in-text citation

Response: The sentence has been revised. The in-text citation has been removed and placed at the end of the sentence.

References 

Comment: Ensure all references are complete and correct

Response: All the reference list has been reworked for completeness and accuracy as indicated in the line numbers

---

## [Decision Letter · Decision Letter 3]

4 Dec 2024

Without a Man's Decision, Nothing Works: Building Resilience to Rift Valley fever in Pastoralist Communities in Isiolo Kenya

PONE-D-24-03443R3

Dear Dr. Mutambo,

We’re pleased to inform you that your manuscript has been judged scientifically suitable for publication and will be formally accepted for publication once it meets all outstanding technical requirements.

Kind regards,

Nasrin Akter, MPH

Guest Editor

PLOS ONE

Additional Editor Comments (optional):

Reviewers' comments:

Reviewer's Responses to Questions

**Comments to the Author**

1. If the authors have adequately addressed your comments raised in a previous round of review and you feel that this manuscript is now acceptable for publication, you may indicate that here to bypass the “Comments to the Author” section, enter your conflict of interest statement in the “Confidential to Editor” section, and submit your "Accept" recommendation.

Reviewer #1: All comments have been addressed

2. Is the manuscript technically sound, and do the data support the conclusions?

Reviewer #1: Yes

3. Has the statistical analysis been performed appropriately and rigorously? 

Reviewer #1: Yes

4. Have the authors made all data underlying the findings in their manuscript fully available?

Reviewer #1: Yes

5. Is the manuscript presented in an intelligible fashion and written in standard English?

Reviewer #1: Yes

6. Review Comments to the Author

Reviewer #1: I have no further comments and recommend this manuscript for publications. Congratulations to the lead author for her efforts throughout the revision process. This is a valuable contribution to the literature base

7. PLOS authors have the option to publish the peer review history of their article (what does this mean?). If published, this will include your full peer review and any attached files.

Reviewer #1: No

---

## [Editor Report · Acceptance letter]

17 Jan 2025

PONE-D-24-03443R3 

PLOS ONE

Dear Dr. Mutambo, 

I'm pleased to inform you that your manuscript has been deemed suitable for publication in PLOS ONE. Congratulations! Your manuscript is now being handed over to our production team.

Kind regards, 

on behalf of

Dr. Nasrin Akter 

Guest Editor

PLOS ONE